# SAMPLE-EFFICIENT LLM OPTIMIZATION WITH RESET REPLAY

## ABSTRACT

Recent advancements in post-training Large Language Models (LLMs), particularly through Reinforcement Learning (RL) and preference optimization methods, are key drivers for enhancing their reasoning capabilities. However, these methods are often plagued by low sample efficiency and a susceptibility to primacy bias, where overfitting to initial experiences degrades policy quality and damages the learning process. To address these challenges, we introduce LLM optimization with Reset Replay (LoRR), a general and powerful plugin designed to enhance sample efficiency in any preference-based optimization framework. LoRR's core mechanism enables training at a high replay number, maximizing the utility of each collected data batch. To counteract the risk of overfitting inherent in high-replay training, LoRR incorporates a periodic reset strategy with reusing initial data, which preserves network plasticity. Furthermore, it leverages a hybrid optimization objective, combining Supervised Fine-Tuning (SFT) and preference-based losses to further bolster data exploitation. Our extensive experiments demonstrate that LoRR significantly boosts the performance of various preference optimization methods on both mathematical and general reasoning benchmarks. Notably, an iterative DPO approach augmented with LoRR achieves comparable performance on challenging math tasks, outperforming some complex and computationally intensive RL-based algorithms. These findings highlight that LoRR offers a practical and sample-efficient paradigm from limited offline data, unlocking greater performance with minimal changes to existing post-training workflows.

## 1 INTRODUCTION

Recent advancements in Large Language Models (LLMs) highlight the potential to enhance reasoning capabilities by leveraging human feedback (Leike et al., 2018). The widely used factor driving this progress is Reinforcement Learning (RL) from human feedback (Ouyang et al., 2022), in which more recent RL approaches assign verifiable rewards (Cui et al., 2025; Ma et al., 2025) based on whether the model's output matches a ground-truth. On the other hand, off-policy methods like Direct Preference Optimization (DPO) (Rafailov et al., 2023) and SimPO (Meng et al., 2024) have streamlined this process by training policies directly from preference data, bypassing the need for an explicit reward model. However, a fundamental limitation of applying these off-policy RL algorithms to real-world tasks is their inefficient utilization of offline datasets. Existing works, such as Meng et al. (2024); Pang et al. (2024), often re-generate responses using base models to maintain proximity to an on-policy setting. When using original datasets directly, a distribution shift between offline datasets and the preference learning process may emerge, making the system susceptible to primacy bias (Nikishin et al., 2022) – a tendency to overfit to initial training experiences. Since the model learns from experiences in static rolled-out datasets rather than through dynamic environmental interactions during training, this early fitting policy can significantly degrade policy quality and hinder learning. In contrast, the SFT algorithm typically trains directly on offline datasets. However, it faces an overfitting problem, and its training is often limited to just a few epochs (Chu et al., 2025), which prevents it from fully utilizing these offline datasets. Consequently, effectively utilizing offline datasets during LLM post-training remains a challenging task.

In traditional RL, a line of work (Nikishin et al., 2022; D'Oro et al., 2023; Xu et al., 2024) addresses the offline datasets learning efficiency challenge by scaling the number of replays, which can remarkably quicken the convergence process and boost the final performance. This finding offers

the understanding that the offline datasets can be utilized more effectively through a greater number of parameter update steps. However, none of the prior works make efforts to enable the updating of LLMs at a *high replay number*, meaning training the LLMs multiple times using the collected experiences for periodic interactions with a dataset. When compared with traditional RL, LLM optimization has parameter updates that occur less frequently. As a result, there is a greater chance in LLM that the offline dataset may not be fully utilized, either by Supervised Fine-Tuning (SFT) or RL-based optimization. If an LLM fails to learn adequately from the current data, it cannot generate new high-quality data, which ultimately hurts its performance.

To address the challenge of effectively utilizing the offline datasets mentioned above, we propose an algorithm called LLM optimization with Reset Replay (LoRR) into finetuning processes. This powerful plugin can enhance the utilization efficiency of offline datasets on any preferred optimization method. LoRR is built upon three essential elements: a high-replay training setting, a reset strategy, and a hybrid optimization technique. First, in contrast to existing on-policy sampling, our algorithm leverages a replay buffer where the sampled experiences are more diverse. This enhanced data diversity is critical for boosting the sample efficiency of fine-tuning by maximizing data reuse with mixing experiences. Capitalizing on this principle, we implement a high-replay-number finetuning process, whereby an LLM is updated multiple times per interactive batch. Second, learning at a high replay number would make LLMs incur a risk of overfitting to the primacy bias, thereby negatively impairing future learning (Nikishin et al., 2022). To overcome this overfitting phenomenon, we introduce a Shrink & Perturb strategy (Ash & Adams, 2020) to periodically reset the network parameters of the LLM, a procedure that preserves the network plasticity critical for continual learning (Sokar et al., 2023; Lyle et al., 2023). Furthermore, given that LoRR updates LLM networks multiple times per batch, we bolster data efficiency by hybridizing SFT and preference-based optimization. Surprisingly, this unique synergy of components achieves a significant advance, enabling LLM finetuning at a replay number an order of magnitude greater than that reported in previous literature.

The contributions of this paper are concluded as follows:

- We provide demonstrations of the existence of the primacy bias in LLM optimization, and show the impact of different experiences on the training process, laying the foundation for mixing rollouts and hybrid optimization.
- We propose a mechanism LoRR for alleviating overfitting and enhancing utilization efficiency of offline datasets, by periodically training with a high replay number and then resetting a part of the LLM;
- Our experiments show that LoRR significantly improves performance when applied to preference methods and can outperform existing complex RL-based algorithms under the same conditions for multi-round improvements.

## 2 RELATED WORKS

**Sample Efficiency for RL**. A central goal in traditional RL is to improve sample efficiency, as acquiring experience through environment interaction is often costly and time-consuming (Chen et al., 2021; D'Oro et al., 2023; Schwarzer et al., 2023). A prominent strategy for achieving this is to maximize the utility of collected data by increasing the replay number, also known as the update-to-data ratio. Despite offering little benefit when applied to standard baselines (Fedus et al., 2020), replay number scaling has been observed to enhance the performance of already well-optimized algorithms. This refers to the number of gradient updates performed on an agent's parameters for each new data point collected (Nikishin et al., 2022; D'Oro et al., 2023). The underlying principle is that an agent can gain more knowledge by repeatedly learning from previously collected experiences, thus reducing the demand for new samples and accelerating learning (Schwarzer et al., 2023). However, while appealing in principle, naively scaling the replay ratio introduces a critical stability issue, overfitting (Nikishin et al., 2022; Lyle et al., 2023; Sokar et al., 2023). From earlier experiences when training, it results in losing the network plasticity to learn good policies in the following learning process. Researchers (D'Oro et al., 2023) further raise the algorithm, using the shrink and perturbation instead of the reset to maintain the network plasticity. While high replay with reset is becoming an active area of investigation in single-/multi-agent RL, this topic remains underexplored within finetuning LLMs, especially as more and more RL algorithms (Rafailov et al., 2023; Cui et al.,

2025) are being applied to LLMs. This gap motivates our work to develop a framework that can harness the benefits of a high replay number while explicitly mitigating the associated risks in the LLM optimization context.

**RL for LLM Reasoning**. RL has been extensively employed in the context of LLMs primarily for aligning models with human preferences (Ouyang et al., 2022; Meng et al., 2024), also in some applications like jailbreaking (Liu et al., 2024). Recently, a growing body of work investigates the amplification of mathematical reasoning in open-source language models, particularly the Qwen2.5 family, through RL (Zeng et al., 2025; Yan et al., 2025; Ma et al., 2025; Cui et al., 2025). Early efforts (Zeng et al., 2025; Ma et al., 2025) employed explicit, verifiable rewards demonstrating notable performance gains on math benchmarks. Subsequent research (Wang et al., 2025b) focused on data efficiency, enabling effective learning even from minimal labeled or unlabeled examples. However, the broader generalizability of these observed gains, particularly within mathematical domains, has faced critical scrutiny. For example, performance enhancements noted on Qwen2.5, even when driven by random or inaccurate rewards, did not consistently translate to other LLMs, suggesting model-specific idiosyncrasies (Zhao et al., 2025). Consequently, while RL may offer a potent tool for specialized LLM enhancement, its successful deployment necessitates a nuanced understanding of its interaction with specific learning dynamics. Our work aims to enhance the sample efficiency of RL to improve the general learning ability of LLMs, extending beyond mere mathematical benchmarks.

**Self-improvement of LLMs** primarily occurs during the training phase, where models enhance their performance by evaluating and refining their own outputs without relying on external human supervision (Kumar et al., 2025). However, a prevailing view suggests that LLMs cannot achieve self-improvement without any external information (Huang et al., 2025). Consequently, an increasing number of methods, especially RL-based, leverage external supervisory signals to facilitate self-improvement, such as outcome labels (Tu et al., 2025; Zhang et al., 2025), value functions (Wang et al., 2024a), and process reward models (Cui et al., 2025). Our approach is based on self-improvement, and it keeps training and partially resets the data and model for each batch.

## 3 THE PRIMACY BIAS IN LLM OPTIMIZATION

This section examines how initial training phases can disproportionately affect the LLM finetuning process due to primacy bias. To start, let us review how previous work (Nikishin et al., 2022; Zhou et al., 2022) defined **the primacy bias** in deep reinforcement learning as follows:

**Definition 1.** *A tendency to overfit initial experiences that damages the rest of the learning process.*

In the context of LLM finetuning, current optimization paradigms often rely on RL or preference-based rewards and collect data via off-policy methods. Typically, a policy periodically rolls out several responses and selects preferences based on the reward verifier. This makes us curious whether there will be primacy bias in finetuning. For this, we present two experiments focused on the LLM finetuning, aiming to illustrate the phenomenon's existence in improper learning from early data. Firstly, we demonstrate that over-training an LLM exclusively on its initial data can irreparably damage the rest of the learning process. Secondly, our experiments reveal that the applicability of the initial experience of LLMs is uncertain during the training process.

**Heavy Priming Causes LLM Overfitting**. The extent to which LLM finetuning relies on its initial data affects its learning process. In order to explicitly reveal the impact of primacy bias in LLM finetuning, we investigate an extreme case of over-reliance on early data: *Does overfitting to just one batch of initial data destroy the generalization of an LLM?* To this end, we design a set of experiments to optimize preferences over *math datasets* based on Llama-3B. As shown in Figure 1, we start by using the default hyperparameters for pure DPO (red line), which means each batch gets one update per step. Then, we set up an identical version of DPO with heavy priming mode (blue line), where we first train the LLM with the first batch 200 times. Only after this extensive initial priming do we resume standard training. The results show that even after training on all the data, the LLM affected by heavy priming was unable to recover from the initial overfitting in the math domain. Besides, both normal training and overfitting have more or less impact on other domain-specific performance, such as health and psychology, suggesting that knowledge of the original model is being forgotten (Ibrahim et al., 2024; Huan et al., 2025). This finding strongly suggests that the primacy bias has compounding effects similar to RL (Nikishin et al., 2022): an LLM that is overfitted early on

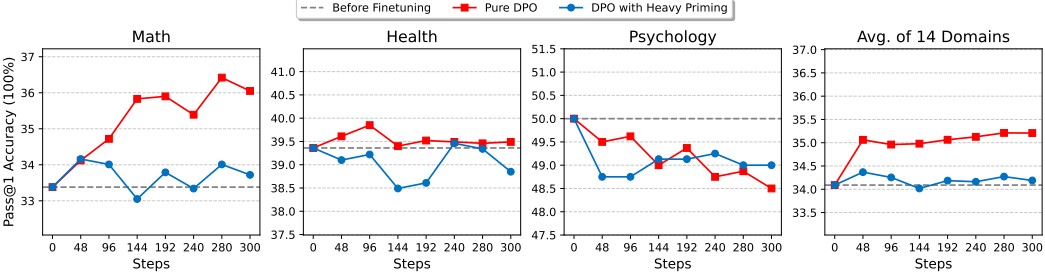

Figure 1: DPO training process gains in MMLU-pro (Avg.) and its subdomains. It compares an LLM trained with standard procedures against one subjected to heavy priming on its initial data.

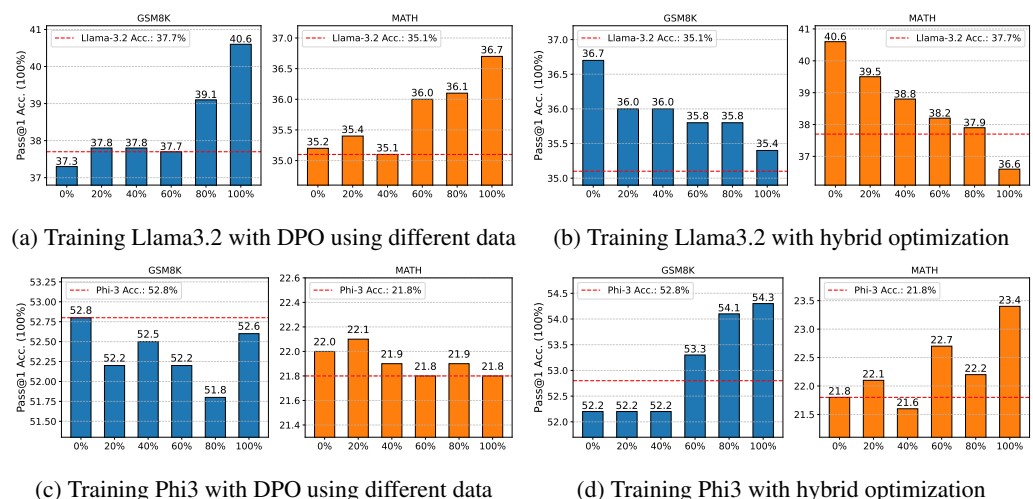

(a) Training Llama3.2 with DPO using different data
(b) Training Llama3.2 with hybrid optimization

(c) Training Phi3 with DPO using different data
(d) Training Phi3 with hybrid optimization

Figure 2: DPO training with different ratios of the rollout data (Figs. (a), (c)) and SFT loss (Figs. (b), (d)), where the higher the percentage the more mixed it is. We pick two benchmarks, GSM8K and Math, to test the models' performance. The red dotted lines are the performance of the base models.

will rollout poorer quality data, which in turn leads to less efficient learning, further impairing its ability to learn iteratively without reset, and so on.

**Initial Experiences are LLM-specific**. Existing preference learning algorithms typically rely solely on roll-out data while ignoring the original responses generated by the base model, which significantly constrains the model's capabilities to the performance bounds of the base model (Ma et al., 2025). However, directly incorporating original data into the optimization process introduces non-trivial challenges: for instance, an LLM cannot initially determine whether the collected original data is useful for learning, leading to potential inefficiencies or even degradation in performance. To explore this issue, we try in practice to optimize LLMs by blending different ratios of data, i.e., combining a portion of the original dataset responses with roll-out preference data scored by a reward verifier. In addition, we integrate diverse optimization losses, such as DPO for preference exploration and SFT for answer fitting. We evaluate two models, Llama3.2 and Phi3, with results shown in Figure 2. Surprisingly, these experiments reveal model-specific patterns rather than consistent trends. Llama3.2, which appears to have acquired strong math domain capabilities during pre-training, leverages roll-out data to generate high-quality samples for further improvement; in contrast, Phi3 requires prior data for SFT fitting, and incorporating initial experience even has a negative impact. These findings highlight that the challenges of integrating original data into preference learning are not universal but LLM-specific, underscoring the need for more thoughtful strategy design to leverage initial experience strategically.

## 4 SCALING REPLAY NUMBER WITH RESET FOR FINETUNING LLM

In reinforcement learning, the term replay number typically refers to the number of agent updates for every single step taken in the environment (Wang et al., 2017; Fedus et al., 2020). Inspired by this,

---

**Algorithm 1** Reset Replay

---

1: **Input:** Language model $\pi_{\theta_{\text{init}}}$, dataset $\mathcal{D}$, reward verifier $r$, sample number $K$, replay number $L$, reset weight $\alpha$, SFT ratio $\lambda_{\text{init}}$, rollout ratio $\varepsilon_{\text{init}}$, and total iteration $N$
2: Initialize policy model $\pi_\theta \leftarrow \pi_{\theta_{\text{init}}}$, reference model $\pi_{\text{ref}} \leftarrow \pi_{\theta_{\text{init}}}$
3: **for** each iteration $n \leftarrow 1$ to $N$ **do**
4:     Sample batch of prompts $\mathcal{B} \sim \mathcal{D}$
5:     **for** each replay $\ell \leftarrow 1$ to $L$ **do**
6:        Adjust rollout ratio $\varepsilon \leftarrow \text{Adj}(\varepsilon_{\text{init}}, \ell)$
7:        Adjust SFT ratio $\lambda \leftarrow \text{Adj}(\lambda_{\text{init}}, \ell)$
8:        $\mathcal{T} \sim \{\}$
9:        **for** each prompt $\mathbf{x} \in \mathcal{B}$ **do**
10:           Generate $K$ responses $\{\mathbf{y}_1, \cdots, \mathbf{y}_K\} \sim \pi_\theta(\cdot|\mathbf{x})$        ▷ annotate preference datasets
11:           Compute outcome rewards $r_i = r(\mathbf{x}, \mathbf{y}_i)$ for $i \in \{1, \cdots, K\}$
12:           Choose the best response $\mathbf{y}_w = \arg\max_{\mathbf{y}_i \in \{\mathbf{y}_1, \cdots, \mathbf{y}_K\}} r_i$,
13:           Choose the worst response $\mathbf{y}_l = \arg\min_{\mathbf{y}_i \in \{\mathbf{y}_1, \cdots, \mathbf{y}_K\}} r_i$,
14:           **if** $\varepsilon < \epsilon$, $\epsilon \sim U(0, 1)$ **then**
15:              Add reannotated samples to $\mathcal{T} \leftarrow \mathcal{T} \cup \{(\mathbf{x}, \mathbf{y}_w, \mathbf{y}_l)\}$        ▷ update buffer
16:           **else**
17:              Add initial samples to $\mathcal{T} \leftarrow \mathcal{T} \cup \{(\mathbf{x}, \mathbf{y}, \mathbf{y}'_l)\}$
18:           **end if**
19:        **end for**
20:        Reset policy model $\pi_\theta \leftarrow \alpha\pi_\theta + (1 - \alpha)\pi_{\theta_{\text{init}}}$        ▷ shrink and perturbation strategy
21:        Update policy model $\pi_\theta$ by hybrid loss on $\mathcal{T}$:
22:        $\mathcal{L}_{\text{SFT}} = -\mathbb{E}_{(\mathbf{x}, \mathbf{y}_w, \mathbf{y}_l) \sim \mathcal{T}} [\log \pi_\theta(\mathbf{y}_w | \mathbf{x})]$
23:        $\mathcal{L}_{\text{DPO}} = -\mathbb{E}_{(\mathbf{x}, \mathbf{y}_w, \mathbf{y}_l) \sim \mathcal{T}} \left[\log \sigma\left(\beta \log \frac{\pi_\theta(\mathbf{y}_w|\mathbf{x})}{\pi_{\text{ref}}(\mathbf{y}_w|\mathbf{x})} - \beta \log \frac{\pi_\theta(\mathbf{y}_l|\mathbf{x})}{\pi_{\text{ref}}(\mathbf{y}_l|\mathbf{x})}\right)\right]$        ▷ loss is flexible
24:        $\mathcal{L}_\theta = \lambda\mathcal{L}_{\text{SFT}} + (1 - \lambda)\mathcal{L}_{\text{DPO}}$
25:        Update reference model $\pi_{\text{ref}} \leftarrow \pi_\theta$
26:     **end for**
27: **end for**
28: **Output:** Optimized policy model $\pi_\theta$

---

we set LLM finetuning algorithms to learn at a high replay number using preference data or verifiable reward. More specifically, the core idea is to update the LLM parameters multiple times for each batch finetuning with the dataset, which is achieved by setting a high replay number while keeping the batch size constant. Surprisingly, despite this seemingly straightforward concept and the amount of work (Nikishin et al., 2022; D'Oro et al., 2023; Schwarzer et al., 2023) that has successfully achieved high replay rates in RL, almost nothing has been reported in the field of LLM finetuning. In fact, both RL and LLM optimization often show that learning without reset leads to a significant primacy bias (Nikishin et al., 2022) and negatively impacts the quality of policies.

To address this issue, we introduce an algorithm called LLM optimization with Reset Replay (LoRR) into finetuning processes. The reset technique we integrated is the Shrink & Perturb strategy (Ash & Adams, 2020), which resets the network parameters of LLMs periodically to maintain the network plasticity. Originally proposed as a method to "warm-start" neural network training for incorporating new data without losing generalization, this technique has recently been adopted in RL systems (D'Oro et al., 2023; Lyle et al., 2023; Xu et al., 2024) to combat overfitting in high replay number settings. The formulation of the reset strategy for LLM optimization is defined as

$$\pi_\theta \leftarrow \alpha\pi_\theta + (1 - \alpha)\pi_{\theta_{\text{init}}}, \tag{1}$$

where $\pi_\theta$ is a current LLM policy learned in the finetuning phase and $\pi_{\theta_{\text{init}}}$ is the initial policy. The interpolation factor $\alpha$ determines the extent to which the existing LLM parameters are preserved during an update. A higher $\alpha$ indicates a greater influence from the new updates, while a lower $\alpha$ means more of the old parameters are retained.

Next, we integrate the details of the above strategy into our proposed LoRR, as shown in Algorithm 1. LoRR follows the general workflow of standard on-policy preference optimization but employs two additional operations. The main operation is in line 20, resetting the policy model $\pi_\theta$ in replay

loops. Given the reset interval $|\mathcal{B}|$ with L times of replay for each batch $\mathcal{B}$, LoRR performs Shrink & Perturb to inject plasticity into the policy model, thereby restoring the learning capability of specified networks. The second is hybrid optimization in line 24 via RL and SFT, which has been shown to work differently for diverse on-policy data (Tajwar et al., 2024; Ma et al., 2025). This augmentation operation with different ratios can take full advantage of LoRR updates, bringing the benefits of diversity to input representations for LLM learning. Below, we'll go over each operation in detail.

**Replays with Reset Data**. As preference algorithms (Meng et al., 2024; Tran et al., 2023) collect data using the on-policy setting, we develop a replay strategy into the reset loop. Specifically, we first sample multiple trajectories $\{\mathbf{y}_1, \cdots, \mathbf{y}_K\}$ for each prompt $\mathbf{x}$, and then re-annotate them with a reward verifier $r$ that selects the highest-scoring one as $\mathbf{y}_w$ and the lowest-scoring one as $\mathbf{y}_l$ (Pace et al., 2024). The preference pairs $(\mathbf{x}, \mathbf{y}_w, \mathbf{y}_l)$ generated by the reward signals are collected in a set of pairs $\mathcal{T}$, a.k.a., a new transition batch. However, to avoid duplicating the update with each replay, a certain ratio of the initial samples is mixed into the transition batch, where the pair is formalized as $(\mathbf{x}, \mathbf{y}, \mathbf{y}_l')$[1]. In this high replay number setting, for every replay $\ell$, we set a linear rollout ratio $\varepsilon$ working on each batch. The ratio goes up as the number of replays declines, which means the model further learns from the experience of the initial sample. In practice, given an initial ratio $\varepsilon_{\text{init}} = 1$, each replay adjusts the ratio to be $\varepsilon \leftarrow \varepsilon_{\text{init}}(1 - \frac{\ell}{2L})$, controlling whether transitions are sampled from the current rolled out sample or the initial sample, to increase the diversity of the transition batch. The intuition behind it is to mitigate experiences of primed LLMs' uncertainty, allowing the model to simultaneously utilize the on/off-policy data (Tajwar et al., 2024), especially at a high replay number, and thus approach the target distribution.

**Replays with Hybrid Optimization**. Both SFT and preference learning methods operate within the same optimal policy-reward subspace, positioning SFT as a special case of implicit reward learning (Wang et al., 2025a). Beyond the preference optimization, RL with SFT has been shown to increase capacity in LLM finetuning (Ma et al., 2025). In §3 we have explored the uncertainty of LLM experience, but what about preference optimization combined with SFT? Thus, in high replay number settings, we directly mix different proportions of SFT losses, allowing the policy model to learn $\mathbf{y}_w$ directly. Mathematically, given a preference loss[2] $\mathcal{L}_{\text{DPO}}$ and an SFT loss $\mathcal{L}_{\text{SFT}}$, the hybrid optimization in each reply is

$$\mathcal{L}_\theta = \lambda \mathcal{L}_{\text{SFT}} + (1 - \lambda)\mathcal{L}_{\text{DPO}}, \tag{2}$$

where $\theta$ denotes the parameter of the current policy $\pi_\theta$ and $\lambda$ is an SFT ratio. The $\mathbf{y}_w$ of the rollout is intuitively consistent with the reward throughout the iteration; hence, as the number of replays increases, so does our $\lambda$. Formally, we set an initial value $\lambda_{\text{init}} = 0$ that the first optimization is based on preferences only, and then for each optimization after a reset, the SFT ratio adjusts as $\lambda \leftarrow \frac{\ell}{2L} + \lambda_{\text{init}}$. As this ratio grows, it was theoretically proven (Tajwar et al., 2024) that the policy distribution slowly aligns with the expected distribution. And thanks to the reset strategy, the hybrid optimization slows down the primary bias in LLM.

The novel combination of the two straightforward yet effective extended techniques allows LLM to not overfit but also to efficiently utilize the diversity of samples. Moreover, LoRR is general and easy to plug into the mainstream off-policy LLM optimization with only minor modification, as detailed in Algorithm 1. Next, we've conducted extensive experiments on various math and reasoning tasks to empirically validate LoRR's effectiveness.

## 5 EXPERIMENTS

In this section, we mainly evaluate LoRR on mathematical and reasoning problems, highlighting the superior performance of LoRR as a plugin with different optimization methods (§5.2). We then investigate the performance of LoRR on iterative DPO and compare it to existing RL-based methods with continuous multi-round improvements, analyzing potential differences in reasoning paradigms (§5.3). Further, we provide an in-depth understanding of the following components: ratio type and replay number in the Appendix §D. For all the tasks, we set $\alpha$ to 0.5 and apply to downstream projections of networks for the Shrink & Perturb strategy.

---

[1]The lose sample $\mathbf{y}_l'$ is from the dataset $\mathcal{D}$ if it have preference data, otherwise, it is rolled out in the initial model $\pi_{\theta_{\text{init}}}$ with the same collection method.

[2]Our default optimizing loss is DPO, but this can be replaced at will, depending on scenarios.

## 5.1 EXPERIMENTAL SETUP

**Datasets**. Our training set is augmented with mathematical problems from a subset of Meta-Math (Yu et al., 2024) and MMIQC (Liu et al., 2025). This data has been processed into high-quality preference data by previous work (Lai et al., 2024) and used for finetuning various mathematical models. To ensure that the amount of data in §5.3 is consistent with the baseline (e.g., DPO-VP (Tu et al., 2025)), the same 8K data are sampled for training. In addition, we optimize on a universal dataset, UltraFeedback (Cui et al., 2023), for comparing the reasoning performance of models.

**Models and Finetuning Implementation**. We first optimize preferences using three model families: Llama3.2-3B (Grattafiori et al., 2024), Phi3-mini-4K (Abdin et al., 2024), and Qwen2.5-3B (Yang et al., 2025), all in the Instruct setup. For a fair comparison, we use the same reward model as SimPO (Meng et al., 2024): ArmoRM-Llama3-8B-v0.1 (Wang et al., 2024b) for ranking generated data, significantly enhancing performance. We test the performance of LoRR under five different sets of preference optimizations: DPO (Rafailov et al., 2023), KTO (Ethayarajh et al., 2024), IPO (Azar et al., 2024), rDPO (Park et al., 2024), and SimPO (Meng et al., 2024), where the hyperparameters remain unchanged and are marked as *Base* in tables. For comparison, we set the number of replays to 3 in LoRR and integrate in these preference optimizations to see the performance of LoRR. Furthermore, we add an iterative training style (*Iter. n*) to each method, where the number of iterations is 3, and each iteration rolls data out through the reward model, so as to compare whether LoRR improves in the case where a policy model meets the same amount of data. For multi-round self-improvement experiments, we mainly use the Qwen2.5-Math-7B model for optimization, which is consistent with most works (Guan et al., 2025; Cui et al., 2025; Zeng et al., 2025).

**Evaluation** In our evaluation, we mainly focus on several well-established mathematical benchmarks: GSM8K (Cobbe et al., 2021), MATH/MATH500 (Hendrycks et al., 2021), TabMWP (Lu et al., 2023), Minerva (Lewkowycz et al., 2022), OlympiadBench (He et al., 2024), and AIME/AMC (Li et al., 2024). All of the benchmarks adopt Pass@1 as the evaluation criterion. Considering reasoning experiments training on UltraFeedback, we further evaluate the generalization ability on MMLU-Pro (Wang et al., 2024c), where we randomize the order of multiple-choice options to avoid contamination.

## 5.2 MAIN RESULTS

Our initial evaluation focuses on quantifying the performance improvement of the LoRR plugin on different preference optimizations. The experimental procedure involves finetuning models on the previously mentioned 8K math dataset, first using standard preference learning to create baseline models, and then enhancing the process with LoRR. Results of these base preference learning and LoRR-based versions for the math ability are shown in Tables 1, 4, and 5. As we can see, the introduction of LoRR, particularly when configured with a replay number of 3, leads to a remarkable boost in performance, far exceeding the outcomes of the base optimization methods. Specifically, the different finetuning models have max 6.54% to 16.99% enhancements relative to the SFT model, using the LoRR setting. It also produces a significant boosting effect compared to methods where the interaction with the dataset is fixed (e.g., the number of iterations is 3). Some optimizations (e.g., training Llama3.2 with rDPO) were originally trained to cause side effects, yet those are higher than SFT models under LoRR. In a side-by-side comparison, SimPO has the most significant performance effect when plugged in LoRR, maintaining first place performance on all three models, followed by rDPO. Note that the small sample size of AIME24 resulted in no significant enhancement, possibly because the SFT data is sampled by $\pi_{\theta_{\text{init}}}$, but this policy has no resolution capability in its knowledge. Overall, LoRR is able to enhance most of the scenarios under different training methods.

To evaluate the broader applicability of LoRR, we extend our experiments to a general domain. We optimize the Llama3.2 on the UltraFeedback dataset and subsequently test on the challenging MMLU-Pro reasoning task. As baselines, we employ two different preference optimization algorithms, DPO and SimPO, and each algorithm incorporates a different contribution of SFT loss. We then augmented these baseline methods with LoRR to isolate its impact. Figure 3 illustrates the comparative performance, highlighting the improvements conferred by our method. Specifically, it show that varying the proportion of SFT loss yield minimal performance improvements and induce overfitting after approximately 400 training steps. In stark contrast, LoRR achieves a significant performance increase by efficiently reusing sample data through its reset-and-replay cycle.

Table 1: Llama3.2 finetuning results on six mathematical benchmarks. We train SFT models on the 8K math data for base, iterative, and LoRR settings, where the training modes include DPO, KTO, IPO, rDPO, and SimPO. For Instruct settings, we use off-the-shelf models as the SFT model.

| Training Method | GSM8k | MATH | TabMWP | Minerva | AMC23 | AIME24 | Avg. |
|---|---|---|---|---|---|---|---|
| Llama3.2-3B-Instruct | 37.7 | 35.1 | 27.5 | 33.8 | 12.5 | _6.7_ | 25.55 |
| Llama3.2 + DPO (Base) | 40.6 | 36.7 | 27.3 | 33.4 | 15.0 | 3.3 | 26.05 |
| + DPO (Iter. n) | 41.9 | 36.1 | 28.1 | 35.2 | 15.0 | _6.7_ | 27.2 |
| + DPO (Base+LoRR) | _43.0_ | 37.9 | 28.2 | 36.0 | 15.0 | **10.0** | 28.35 |
| Llama3.2 + KTO (Base) | 39.0 | 36.1 | 27.3 | 35.2 | _17.5_ | _6.7_ | 26.97 |
| + KTO (Iter. n) | 39.4 | 36.2 | 26.8 | 32.4 | **20.0** | **10.0** | 27.47 |
| + KTO (Base+LoRR) | 40.5 | 37.3 | 28.8 | 36.2 | **20.0** | **10.0** | 28.80 |
| Llama3.2 + IPO (Base) | 37.8 | 36.0 | 26.7 | 33.6 | 15.0 | _6.7_ | 25.97 |
| + IPO (Iter. n) | 40.9 | 36.9 | 27.7 | 35.4 | 12.5 | _6.7_ | 26.70 |
| + IPO (Base+LoRR) | 40.3 | 37.1 | 28.9 | 36.0 | 12.5 | _6.7_ | 26.92 |
| Llama3.2 + rDPO (Base) | 37.7 | 35.9 | 27.6 | 33.4 | 15.0 | 3.3 | 25.48 |
| + rDPO (Iter. n) | 40.6 | 37.8 | _29.5_ | _36.6_ | 12.5 | _6.7_ | 27.28 |
| + rDPO (Base+LoRR) | 40.9 | _38.1_ | 28.4 | **39.4** | 20.0 | _6.7_ | _28.92_ |
| Llama3.2 + SimPO (Base) | 39.0 | 36.8 | 28.4 | 33.8 | 12.5 | 3.3 | 25.63 |
| + SimPO (Iter. n) | 40.9 | 37.6 | 26.4 | 34.4 | 15.0 | _6.7_ | 26.83 |
| + SimPO (Base+LoRR) | **45.7** | **38.6** | **32.3** | **39.4** | **20.0** | 3.3 | **29.88** |

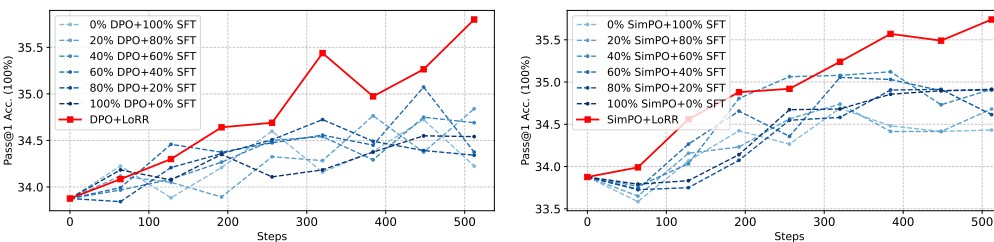

(a) Training with DPO on UltraFeedback  (b) Training with SimPO on UltraFeedback

Figure 3: Average pass@1 accuracy of fine-tuned LLama3.2 models on MMLU-Pro's 14 domain test benchmarks. Since different replays of LoRR change the SFT ratio, we select preference learning to mix varying ratios of SFT losses as baselines.

## 5.3 RESULTS ON ITERATIVE DPO

Further, to create a rigorous benchmark against RL and preference learning methods, we conduct iterative DPO (a.k.a., multi-round DPO) on the advanced Qwen2.5-Math-7B model (Yang et al., 2024), ensuring a controlled comparison by using the same base model and datasets with similar sample sizes. We implement the LoRR with 3 epochs of DPO. At the end of each epoch, we dynamically generate new SFT and preference data using the current policy $\pi_\theta$. This newly generated preference data is labeled by a reward model, making it closer to an on-policy setting, as discussed in (Meng et al., 2024). The resulting policy, which we name it as Qwen2.5-7B-DPO-LoRR, is then compared against nine state-of-the-art RL-based methods that introduce in Appendix C.

These baselines all use the ZERO version, so all answers **y** to the maths questions are rolled out by policies not initially in the dataset. We evaluate our model and baselines on GSM8K (Cobbe et al., 2021), MATH (Hendrycks et al., 2021), and several Olympiad-level benchmarks (He et al., 2024), including AMC23, AIME24, and OlympiadBench. However, most of these have different training data and reward models, making it difficult to make fair comparisons. To align with most RL methods, we similarly use the 8K math questions we mentioned to train our model with iterative DPO. Table 2 lists the pass@1 accuracies of different models, where the first category is not comparable due to the effect of the amount of training data, etc., and the second category is relatively fair, although the different methods differ in their settings, e.g., the data and epoch. We also present Table 6 for further analysis of training data consumption and GPU usage to describe the difference among them. Note that we do not use rule-based reward signals because LoRR can train on general datasets, not just for answer matching.

Table 2: Pass@1 results across different methods on math tasks. All the models have been fine-tuned based on the Qwen2.5-Math-7B. The results with † are the ones we evaluated from the official model published by the authors, and the rStar results with ∗ are obtained from Guan et al. (2025). In the relatively fair comparison, **Bold** indicates the best results and underline indicates the second one.

| Model Pass@1 Acc. | MATH500 | Minerva | Olybench | AMC23 | AIME24 | **Avg.** |
|---|---|---|---|---|---|---|
| Qwen2.5-Math-7B† | 66.2 | 10.7 | 24.1 | 47.5 | 23.3 | 34.4 |
| Qwen2.5-Math-7B-Instruct† | 84.6 | 37.1 | 39.9 | 62.5 | 16.7 | 48.2 |
| rStar-Math-7B∗ (Guan et al., 2025) | 78.4 | - | 47.1 | 47.5 | 26.7 | - |
| Eurus-2-7B-PRIME† (Cui et al., 2025) | 74.0 | 39.7 | 35.6 | 57.5 | 23.3 | 46.0 |
| Qwen2.5-7B-ReLIFT† (Ma et al., 2025) | 81.4 | 23.9 | 44.3 | 62.5 | 13.3 | 45.1 |
| Qwen2.5-7B-LUFFY† (Yan et al., 2025) | 80.6 | 30.5 | 45.6 | 72.5 | 13.3 | 48.5 |
| Simple-RL-Zero† (Zeng et al., 2025) | **77.8** | 32.7 | **40.7** | 57.5 | 20.0 | 45.7 |
| Qwen2.5-7B-PURE-VR† (Cheng et al., 2025) | 72.8 | 13.6 | 28.1 | 50.0 | 23.3 | 37.6 |
| Qwen2.5-DPO-R1-Zero† (Zhang et al., 2025) | 75.4 | 27.6 | 38.8 | 60.0 | 23.3 | 45.0 |
| Qwen2.5-7B-DPO-VP† (Tu et al., 2025) | 76.0 | 30.9 | 38.2 | **62.5** | 26.7 | 46.9 |
| Qwen2.5-7B-DPO-LoRR-iter1 | 73.8 | 23.5 | 35.1 | 55.0 | 23.3 | 42.1 |
| Qwen2.5-7B-DPO-LoRR-iter2 | 75.4 | 33.8 | 37.0 | 55.0 | **30.0** | 46.2 |
| Qwen2.5-7B-DPO-LoRR-iter3 | 75.6 | **36.0** | 37.2 | 60.0 | **30.0** | **47.8** |

Based on the results in Table 2 and Table 6 (in Appendix C), we draw the following conclusions:

- **Trained with LoRR, a simple iterative DPO achieves mathematical reasoning that is comparable to RL-based methods.** Our final model, Qwen2.5-7B-DPO-LoRR, attains an average score of 47.8, the highest among all methods in the relatively fair comparison group. This performance not only exceeds other DPO variants like Qwen2.5-7B-DPO-VP (46.9) but also outperforms prominent RL-based models such as Simple-RL-Zero (45.7), all while using the same base model and initial training data constraints.

- **The iterative application of DPO demonstrates consistent and significant performance improvements.** We observe a clear monotonic progression in the average score across iterations: from 42.1 in the first iteration, to 46.2 in the second, and culminating in 47.8 in the third. This highlights the effectiveness of our reset replays for data reusing.

- **This superior performance is achieved with remarkable data and computational efficiency, presenting a highly practical and scalable paradigm.** Our method requires only 8K preference pairs for the entire iterative training process, and the entire training for our model was completed in just 2 days on a single A100 GPU. This is a fraction of the data used by other top-performing models like Qwen2.5-7B-ReLIFT (45K) and orders of magnitude less than rStar-Math-7B (∼3.6M) with more GPUs, demonstrating our approach's ability to learn effectively from limited data.

In summary, our work demonstrates that LoRR plugin an iterative DPO can achieve leading performance in mathematical reasoning without the need for massive-scale SFT data, complex reward modeling pipelines, or extensive computational clusters, offering a powerful and accessible alternative to prevailing RL-based finetuning methodologies.

# 6 CONCLUSION

In this paper, we introduced the LLM optimization with Reset Replay (LoRR), to address the critical issues of low sample efficiency and primacy bias in LLM fine-tuning. Our method is a general-purpose plugin that synergizes high-replay training with periodic parameter resets and a hybrid optimization loss, enabling more effective data utilization. Extensive experiments show that LoRR significantly enhances various preference optimization methods on both mathematical and general reasoning tasks. Notably, an iterative DPO approach augmented with LoRR achieves state-of-the-art performance, outperforming complex RL-based methods with substantially greater sample and computational efficiency. This work demonstrates that by mitigating overfitting from intensive data reuse, LoRR unlocks the latent potential in training data, offering a practical path toward more powerful and efficient LLM optimization. As for future work, there are two key aspects. On one hand, a higher replay number (e.g., $L = 10$) tends to become less effective in the later stages of training, therefore, it may be feasible to address this issue through adaptive replay frequencies. On the other hand, it is crucial to further understand and reveal the potential mechanisms of plasticity in LLM networks when they are fine-tuned.

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

Table 3: Various preference optimization hyperparameters used for each training setting.

| Method | $\beta$ | $\gamma$ | Learning rate |
|--------|------|------|---------------|
| DPO | 0.01 | - | 5.0e-7 |
| KTO | 0.01 | 1.0 | 5.0e-7 |
| IPO | - | 0.5 | 5.0e-7 |
| rDPO | 0.01 | 0.6 | 5.0e-7 |
| SimPO | 2 | 0.55 | 1e-6 |

Table 4: Phi3 finetuning results on six mathematical benchmarks. We train SFT models on the 8K math data for base, iterative, and LoRR settings, where the training modes include DPO, KTO, IPO, rDPO, and SimPO. For Instruct settings, we use off-the-shelf models as the SFT model.

| Training Method | GSM8k | MATH | TabMWP | Minerva | AMC23 | AIME24 | Avg. |
|-----------------|-------|------|--------|---------|-------|--------|------|
| Phi3-mini-4K-Instruct | 52.8 | 21.8 | 19.2 | 23.4 | 20.0 | 3.3 | 23.42 |
| Phi3 + DPO (Base) | 52.6 | 21.8 | 18.7 | 24.4 | 12.5 | 3.3 | 22.22 |
|     + DPO (Iter. n) | 51.9 | 19.9 | 17.2 | 18.2 | 22.5 | 3.3 | 22.17 |
|     + DPO (Base+LoRR) | 53.9 | 23.8 | 17.6 | **25.2** | 15.0 | **6.7** | 23.70 |
| Phi3 + KTO (Base) | 52.8 | 21.6 | 19.0 | 23.8 | 17.5 | 3.3 | 23.00 |
|     + KTO (Iter. n) | 51.3 | 21.1 | 18.9 | 18.2 | 20.0 | 3.3 | 22.13 |
|     + KTO (Base+LoRR) | **54.2** | 23.8 | 19.7 | 24.2 | 20.0 | 3.3 | 24.20 |
| Phi3 + IPO (Base) | 49.7 | 20.2 | 16.6 | 18.4 | 15.0 | 3.3 | 20.53 |
|     + IPO (Iter. n) | 51.9 | 21.1 | 19.5 | 20.2 | 20.0 | 3.3 | 22.67 |
|     + IPO (Base+LoRR) | 51.9 | 21.7 | 19.7 | 19.8 | 17.5 | **6.7** | 22.88 |
| Phi3 + rDPO (Base) | 52.8 | 21.7 | **19.8** | 20.6 | 20.0 | 0.0 | 22.48 |
|     + rDPO (Iter. n) | 50.2 | 20.6 | 17.7 | 19.8 | 17.5 | 3.3 | 21.52 |
|     + rDPO (Base+LoRR) | 52.5 | 22.4 | 18.1 | 23.6 | **25.0** | 3.3 | 24.15 |
| Phi3 + SimPO (Base) | 51.7 | 21.2 | 19.5 | 22.2 | 17.5 | 3.3 | 22.57 |
|     + SimPO (Iter. n) | 48.7 | 19.2 | 16.1 | 18.6 | 17.5 | **6.7** | 21.13 |
|     + SimPO (Base+LoRR) | 53.2 | **24.2** | 19.0 | 24.0 | 22.5 | **6.7** | **24.93** |

## A  IMPLEMENTATION DETAILS

In this section, we outline the specific parameters and data of the experiments.

**General training hyperparameters**. We conduct preliminary experiments for the basic preference learning settings using a batch size of 128 and a single training epoch. Moreover, we set the maximum sequence length at 2048 and utilize the AdamW optimizer with a cosine learning schedule, which includes 10% warmup steps, for the preference optimization. For each preference method, we follow the most parameterised exploration of SimPO (Meng et al., 2024), where Table 3 shows the detailed hyperparameters for baselines. For LoRR, we set the default parameters: a replay number $L = 3$, sample number $K = 5$, reset weight $\alpha = 0.5$, and an iteration number $N = 7$ for math. For LoRR in iterative DPO, $N = 3$ is set and 3 epochs are guaranteed, while other parameters remain unchanged. DeepSpeed ZeRO2 with CPU offload is used to reduce GPU memory usage during training.

**Building datasets for optimization**. For math tasks, our training dataset is supplemented with mathematical problems selected from subsets of Meta-Math (Yu et al., 2024) and MMIQC (Liu et al., 2025). To maintain consistency in data volume with the baselines as discussed in §5.3, we sampled the same 8K data points for training. For the reasoning tasks, we conduct optimizations on a general-purpose dataset, UltraFeedback (Cui et al., 2023), to facilitate comparisons of the models' reasoning capabilities. Regarding training data, we produce a maximum of 5 responses for each prompt in each replay. Additionally, we leverage the ArmoRM model (Wang et al., 2024b) to label the preference relationships between these responses.

Table 5: Qwen2.5 finetuning results on six mathematical benchmarks. We train SFT models on the 8K math data for base, iterative, and LoRR settings, where the training modes include DPO, KTO, IPO, rDPO, and SimPO. For Instruct settings, we use off-the-shelf models as the SFT model.

| Training Method | GSM8k | MATH | TabMWP | Minerva | AMC23 | AIME24 | **Avg.** |
|---|---|---|---|---|---|---|---|
| Qwen2.5-3B-Instruct | 41.1 | 60.4 | 32.1 | 60.8 | 25.0 | 3.3 | 37.12 |
| Qwen2.5 + DPO (Base) | 44.3 | 60.9 | 32.4 | 60.2 | 25.0 | 0.0 | 37.13 |
| + DPO (Iter. n) | 42.5 | 60.8 | 30.5 | 60.0 | 27.5 | 0.0 | 36.88 |
| + DPO (Base+LoRR) | **45.4** | **61.1** | 32.5 | 61.2 | 27.5 | 0.0 | 37.95 |
| Qwen2.5 + KTO (Base) | 44.3 | 60.5 | 29.6 | 60.6 | 22.5 | 0.0 | 36.25 |
| + KTO (Iter. n) | 43.1 | 60.6 | 32.3 | 60.2 | 27.5 | 0.0 | 37.28 |
| + KTO (Base+LoRR) | 44.3 | 59.8 | 30.9 | 60.4 | 27.5 | 3.3 | 37.70 |
| Qwen2.5 + IPO (Base) | 43.4 | 60.3 | 31.2 | 61.4 | 22.5 | 3.3 | 37.02 |
| + IPO (Iter. n) | 42.3 | 60.0 | 28.1 | 60.2 | 30.0 | 0.0 | 36.77 |
| + IPO (Base+LoRR) | 44.0 | 60.3 | 31.1 | **61.8** | 27.5 | **6.7** | 38.57 |
| Qwen2.5 + rDPO (Base) | 44.5 | 60.8 | 29.6 | 61.4 | 22.5 | 3.3 | 37.02 |
| + rDPO (Iter. n) | 43.1 | 60.6 | 32.3 | 61.2 | 27.5 | 0.0 | 37.45 |
| + rDPO (Base+LoRR) | 44.9 | **61.1** | 30.3 | 58.0 | 32.5 | 3.3 | 38.35 |
| Qwen2.5 + SimPO (Base) | 41.8 | 60.6 | 29.6 | 61.0 | 27.5 | 3.3 | 37.30 |
| + SimPO (Iter. n) | 37.7 | 57.4 | 28.4 | 58.6 | 32.5 | **6.7** | 36.88 |
| + SimPO (Base+LoRR) | 41.8 | 60.4 | **32.9** | **61.8** | **35.0** | **6.7** | **39.77** |

## B  ADDITIONAL EXPERIMENT RESULTS

We further evaluate the performance of the LoRR plugin on Phi3 and Qwen2.5 models, with results presented in Tables 4 and 5 respectively. These experiments follow the same setup as the Llama3.2 evaluation: finetuning on the 8K math dataset using standard preference learning (baseline) and LoRR-enhanced preference learning, with performance measured across the six mathematical benchmarks. Similarly to the body text, the LoRR-based versions consistently outperform their base preference learning and iterative counterparts. The results for Phi3 and Qwen2.5 reinforce the effectiveness of LoRR across different model architectures, consistently enhancing performance across most mathematical benchmarks when integrated with various preference optimization techniques.

## C  SUPPLEMENTS TO ITERATIVE DPO

Here, we introduce some baselines based on RL and iterative DPO methods for our comparison, and compare their efficiency in learning offline data.

- Qwen2.5-Math-7B-Instruct (Yang et al., 2024): The instruction-tuned base model from the Qwen family, selected for its strong mathematical reasoning capabilities.

- rStar-Math-7B (Guan et al., 2025): A model trained on a large volume of data generated via self-evolution with Monte Carlo Tree Search, which is then filtered using a Process Preference Model.

- Eurus-2-7B-PRIME (Cui et al., 2025): A model trained using policy rollouts and outcome labels, leveraging implicit process rewards.

- Simple-RL-Zero (Zeng et al., 2025) and Qwen2.5-7B-PURE-VR (Cheng et al., 2025): Two similar models, trained by RL with verifiable rewards and some tricks, were also trained without additional SFT data.

- Qwen2.5-7B-ReLIFT (Ma et al., 2025) and Qwen2.5-7B-LUFFY (Yan et al., 2025): Two models are both reinforcement learning frameworks that bridge the gap between RL and SFT by incorporating off-policy reasoning traces into the training process.

- Qwen2.5-DPO-R1-Zero (Zhang et al., 2025) and Qwen2.5-7B-DPO-VP (Tu et al., 2025): Both investigate the effectiveness of iterative DPO in facilitating self-improvement for LLMs.

Table 6: A comparison of data and GPUs among different methods.

| | Qwen2.5-Math-7B-Instruct | rStar-Math-7B | Eurus-2-7B-PRIME | Qwen2.5-7B-SimpleRL-Zero | Qwen2.5-7B-ReLIFT | Qwen2.5-7B-DPO-VP | Qwen2.5-7B-DPO-LoRR |
|---|---|---|---|---|---|---|---|
| **Paradigm** | RL + ORM | MCTS + PPM | RL + PRM | RL + VR | RL + SFT | DPO + VR | DPO + SFT |
| **Base Model** | Qwen2.5-Math-7B | Qwen2.5-Math-7B | Qwen2.5-Math-7B | Qwen2.5-Math-7B | Qwen2.5-Math-7B | Qwen2.5-Math-7B | Qwen2.5-Math-7B |
| **SFT Data** | 2.5M (open-source and in-house) | ~7.3M (MATH, NuminaMath, etc.) | 230K | 0 | 0 | 0 | 0 |
| **RM Data** | 618K (in-house) | ~7K (in-house) | 0 | 0 | 0 | 0 | 0 |
| **RM** | Qwen2.5-Math-RM (72B) | 7B PPM | Eurus-2-7B-SFT | Rule-based RM | Rule-based RM | DeepSeek-V3 | ArmoRM-Llama3-8B-v0.1 |
| **Improvement Data** | 66K | ~3.647M | 150K | 8K | 45k | 8K | 8K |
| **Epoch** | - | 2 | - | 1 | 3 | 6 | 3 |
| **GPUs / Time** | - | 80 H100 / few days | 8 A100 / few days | 32 H100 / 1.5 days | 16 A800 / - | 1 A100 / 3 days | 1 A100 / 2 days |

# D  ABLATION STUDY AND ANALYSIS

To further analyze the components of LoRR, we conduct an ablation study to assess the effectiveness of our LoRR. Specifically, we first compare the following three ablation settings for training DPO: (i) replays only with reset data, i.e., we adjust the rollout ratio in LoRR; (ii) replays only with hybrid optimization (adjusting SFT ratio) while the training data is all from the policy rollouts; and (iii) a full LoRR algorithm with both components. For comparison, we use a pure DPO as a baseline. As summarized in Figure 4, the results show that both components more or less enhance the DPO performance. Specifically, replays with reset data require high-quality model rollouts, so performance depends heavily on the base versus base models. Hybrid optimization, on the other hand, substantially improves sample efficiency in reset replays, resulting in better performance.

We also explore the effect of a high number of replays, visualizing the performance of LoRR under different $L$ values on the six math benchmarks. As presented in Figure 5, we observe that $L \in \{2, 3, 5\}$ could drastically improve the sample efficiency. This conclusion is similar to traditional RL agents (D'Oro et al., 2023; Xu et al., 2024), but we validate it for the first time in a language task. However, a larger replay number, such as $L = 10$, causes poor performance. A possible reason for this decline is that overly frequent exploitation of the collected data can lead to overfitting on the current experience, even with the reset. This, in turn, causes the policy to quickly lose its ability to sample high-quality rollouts. Thus, considering the training time and sample efficiency, we choose $L = 3$ as the benchmark setting for LoRR.

Consequently, these results affirm that current preference learning does not effectively utilize the data it gathers. LoRR, conversely, can activate its latent potential as an efficient replay strategy.

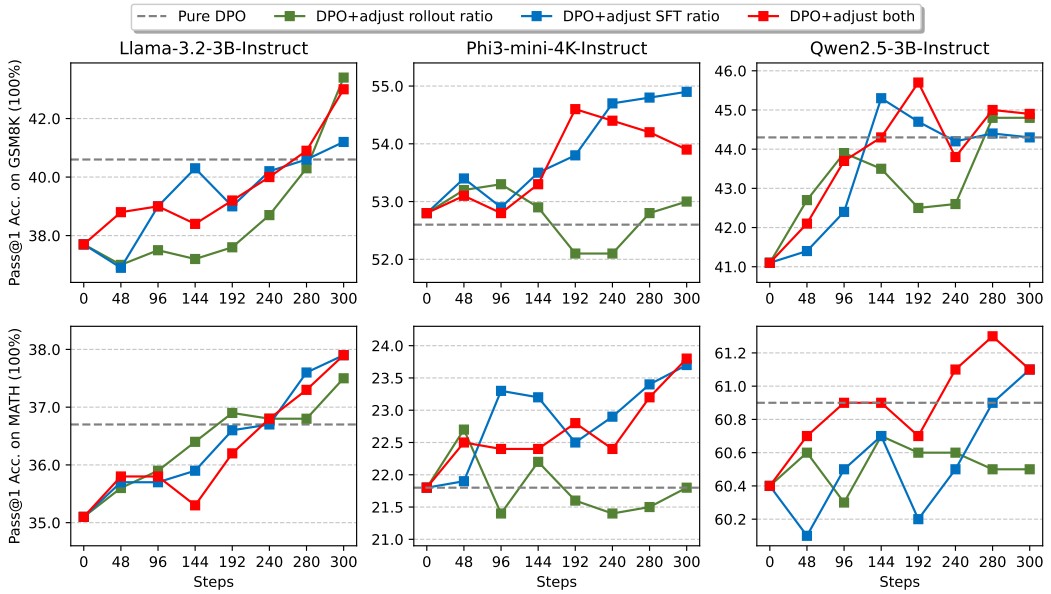

Figure 4: Comparison of different components of LoRR. The experiments were fine-tuned on each of the three language models with DPO, and tested on GSM8K and MATH tasks.

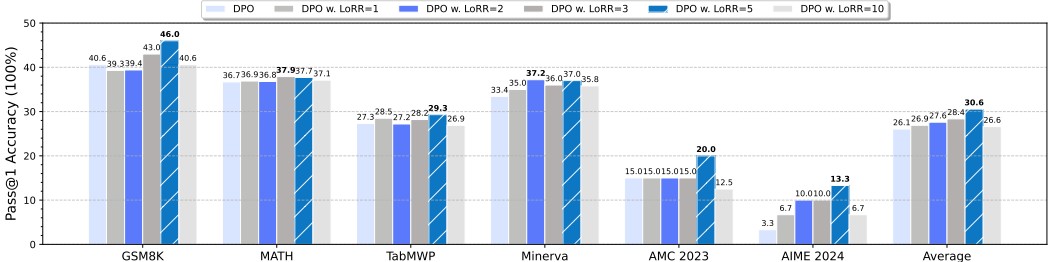

Figure 5: The performance of DPO with LoRR on six math tasks under a different replay number. The best results are described in **bold**, and the average of the six tasks is shown at the end.