# OpenReview forum: "Sample-efficient LLM Optimization with Reset Replay"
_ICLR.cc/2026/Conference — Submitted to ICLR 2026_

### Official Review · Reviewer_zNtC · 2025-10-27

**Soundness:** 2
**Presentation:** 2
**Contribution:** 1
**Rating:** 2
**Confidence:** 4

**Summary:**

This manuscript introduces 'LLM Optimization with Reset Replay' (LoRR), a plugin method designed to enhance preference-based fine-tuning of Large Language Models (LLMs). The core idea is to enable high-intensity training on a single data batch by combining a high replay number with periodic parameter resets and a hybrid optimization objective. While the individual components of LoRR are derived from existing concepts in machine learning. The experimental evidence presented is compelling, demonstrating remarkable computational efficiency compared to standard methods and even complex RL-based algorithms.

**Strengths:**

1. The LoRR method is mechanically simple and well-motivated.
2. It provides a easy-to-implement plugin that delivers substantial performance gains.

**Weaknesses:**

1. The claim of novelty requires careful qualification. The manuscript does not introduce a fundamentally new algorithmic concept. The Reset Replay, iterative DPO was proposed before. The core contribution lies in the  application of established techniques to a specific, high-impact problem domain.
2. The experiment results are not enough to prove its effectiveness. In Table 2, the performance of Qwen2.5-7B-DPO-LoRR is not better than Qwen2.5-7B-DPO-LoRR or those SFR+RFT methods [1,2].
3.  The use of fixed hyperparameters (L, α) is a clear limitation.  The author should focus on developing adaptive schedulers that can dynamically adjust the replay number and reset intensity based on training state.
4. The comparsion with pure SFT methods using less expert data (such as LIMO，S1) should be given. The efficiency of the proposed method and iterative DPO methods should be compared in detail.

**Questions:**

please see the weaknesses.

---

### Official Review · Reviewer_gP68 · 2025-10-30

**Soundness:** 3
**Presentation:** 3
**Contribution:** 3
**Rating:** 6
**Confidence:** 2

**Summary:**

This paper addresses the low sample efficiency and primacy bias issues in post-training optimization of Large Language Models. It proposes a general plugin framework called LoRR, which integrates three core mechanisms: high-replay training, periodic parameter reset based on the Shrink and Perturb strategy, and hybrid optimization combining Supervised Fine-Tuning and preference-based losses. By maximizing the reuse of offline data batches and mitigating overfitting caused by high-replay training, LoRR aims to enhance the performance of preference-based optimization methods. Extensive experiments on mathematical reasoning and general reasoning benchmarks show that LoRR can significantly improve the performance of various preference optimization methods and enable iterative DPO to achieve performance comparable to complex RL-based algorithms with higher sample and computational efficiency.

**Strengths:**

1. Targeted solution to key LLM optimization challenges: LoRR effectively addresses two critical pain points in LLM post-training: low sample efficiency of offline data utilization and primacy bias leading to early overfitting. Unlike existing methods that either re-generate data to maintain on-policy characteristics or are limited by SFT's few-epoch training, LoRR achieves efficient data reuse through high-replay training while preserving network plasticity via periodic resets, filling the gap in efficient offline data utilization for LLM optimization.
2. Strong generality and practicality: As a plug-and-play framework, LoRR can be seamlessly integrated into mainstream preference-based optimization methods without requiring major modifications to existing post-training workflows. Experiments on multiple model families (Llama3.2, Qwen2.5) demonstrate its stable effectiveness across different architectures, making it easily applicable to existing LLM optimization pipelines and enhancing its industrial application value.

**Weaknesses:**

1. Limited generalization across model scales: LoRR’s experiments are restricted to models with 3B-7B parameters. Larger models typically have stronger resistance to overfitting and different network plasticity, this limitation makes it impossible to confirm whether LoRR maintains effectiveness across diverse model sizes.
2. The reset strategy relies on a fixed interpolation factor α , but the paper does not analyze how α affects performance across different datasets or tasks. For example, whether sparse datasets require smaller α to retain more initial parameters, or whether complex reasoning tasks need larger α to accept more new updates. Additionally, the replay number L is fixed at 3, and no adaptive adjustment mechanism is proposed. This one-size-fits-all hyperparameter setting may limit LoRR's adaptability to diverse task scenarios.

**Questions:**

Please see Weaknesses above.

---

### Official Review · Reviewer_rXRd · 2025-10-30

**Soundness:** 3
**Presentation:** 3
**Contribution:** 2
**Rating:** 4
**Confidence:** 5

**Summary:**

The paper argues that preference-based LLM post-training suffers from a primacy-style effect where very early data dominates the rest of training. It proposes LoRR, which keeps replaying the same batch several times, then interpolates the model back toward the initializer and increases an SFT term so the model does not get stuck. When LoRR is attached to several preference losses, the paper reports steady gains on math and reasoning benchmarks using about 8k preference data.

**Strengths:**

1.  The problem setup is timely. The paper connects the low sample-efficiency of preference post-training to a familiar RL idea, namely that early offline data can dominate, so the motivation is easy to read.
2.  The method is simple to plug in. It is basically higher replay, a shrink and perturb style reset, and a replay-aware SFT mix, and it is written to work with multiple preference optimizers rather than a single one.
3.  On the intended low-data math setup, the LoRR variants are consistently above the corresponding base preference learners and also above the iterative baselines listed in the appendix, which fits the paper’s story.
4.  The paper explains LoRR using ideas from high replay in RL, so it does not look like a collection of unrelated tricks.

**Weaknesses:**

1.  The algorithm in Sec. 4 is not tight enough for someone to reimplement without guessing. In Algorithm 1 the condition on the rollout ratio is written in a way that flips the intended Bernoulli test, the triple $(x, y, y'_l)$ is inserted without saying what $y$ is, and the reset is shown inside the replay loop even though the text elsewhere calls it periodic.
2.  Figure 5 does contain a replay ablation and it is good to see that. It shows that replay counts 2, 3 and 5 improve the average over six math tasks, while 10 makes things worse. So the data actually say there is a useful middle range and that pushing replay too far harms rollout quality, which is a familiar RL pattern. The introduction, however, still talks about enabling a replay number that is much higher than before, and this does not really match the curve in Figure 5, because the curve peaks around 3 to 5 and then drops. It would help to explain why the paper fixes on ($L=3$) even though ($L=5$) also looks strong.
3.  Important hyperparameters are kept fixed. In Sec. 5 they simply set ($\alpha=0.5$) and apply the reset to downstream projections, but there is no experiment showing what happens if ($\alpha$) is smaller or if more layers are reset or if the SFT cap is different. Since the purpose of LoRR is to avoid overfitting while replaying, this makes the method look more sensitive than the text suggests.

**Questions:**

1.  What is the exact reset schedule we should follow in code? Algorithm 1 resets inside each replay loop, the text in Sec. 4 describes the reset as periodic, and the experiments in Sec. 5 say you apply it to downstream projections with ($\alpha=0.5$). It would help if you could specify the frequency (per replay, per batch, per N steps) and the exact parameter groups.
2.  Figure 5 shows a clear pattern. Replays 2, 3 and 5 are good, but 10 is bad. Can you explain the mechanism for the drop at 10? Is it that rollouts get worse because the model keeps reusing slightly off-policy data, or is it interaction with the hybrid loss that becomes too SFT-heavy, or simply that the same batch gets seen too many times? Also, do you expect the optimal replay count to change for tasks that are not math?
3.  Section 5.1 says you also include an iterative training style (Iter. n) for each preference method, with 3 iterations and re-rolling through the reward model, and Tables 4 and 5 report Iter. n numbers for Qwen2.5 and Llama3.2. Could you spell out the exact loop for Iter. n? For example, do you regenerate preferences from the current policy at every iteration, do you keep the same 8k pool and only update scores, and do you also reset between iterations? Right now Iter. n is used as a baseline but it is not described at the same level of detail as LoRR.
4.  In Figure 1 you make the central point that heavy priming on the first batch damages later learning and also affects other MMLU-Pro domains. Do you believe the same conclusion holds for long-thinking models that already generate multi-step chains and may redistribute attention across steps? It would be useful to know whether the primacy effect you saw with DPO on Llama-style models transfers to models that run long, more general reasoning patterns.
5.  The SFT weight increases with the replay index and in the example it tops out at 0.5. Was that chosen to prevent the preference signal from being drowned out, or was it just a convenient value? If you let it reach 1.0 on the last replay, do you get better exploitation, or does the model collapse faster?
6.  Since LoRR explicitly reuses the same preference data more times, how robust is it to slightly noisy or biased preferences, for example if the reward model is a few points worse than ArmoRM? A small noised labels experiment would make the sample efficiency claim stronger.

---

### Official Review · Reviewer_QvM5 · 2025-10-30

**Soundness:** 3
**Presentation:** 3
**Contribution:** 3
**Rating:** 4
**Confidence:** 3

**Summary:**

The paper proposes LoRR: a plug-in that (1) increases the replay/update-to-data ratio during preference-based finetuning, (2) periodically shrinks & perturbs (interpolates) the model toward its initialization to restore plasticity, and (3) mixes SFT and preference losses to improve data reuse; experiments report substantial gains on math and reasoning benchmarks.

**Strengths:**

1. LoRR can be integrated with multiple preference optimization algorithms and shows particular advantages in low-compute scenarios.
2. The paper comprehensively evaluates LoRR on three model families across both math reasoning and general domains.

**Weaknesses:**

1. The method feels like a combination of three previously known method, without a unifying theoretical explanation or deeper empirical analysis of why they can work together.
2. The example  in Figure 1 showing “DPO with heavy priming” (training on the same data 200 times) seems unrealistic—no one would actually perform 200 passes over the same dataset. Moreover, the appendix itself reports that L=10 already causes degradation, making the L=200 example appear artificially extreme. A fairer comparison would be: 100% data with 1× training, 50% data with 2× training, 25% data with 4× training, and 12.5% data with 8× training, which would more meaningfully test data-reuse efficiency.

**Questions:**

1. The main paper should include more ablation and analysis experiments to isolate their individual contributions. Please also explore hyperparameter sensitivities—e.g., how $\lambda$ and $\alpha$ schedules or different initialization choices affect performance.
2. In Algorithm 1, line 17, the notation $y^′_l$ is unclear.
3. How does the proposed method scale to larger model sizes? It would be useful to include results or discussion on its generalization behavior for models beyond 7B parameters.

---

### Meta-Review · Area_Chair_BbLG · 2025-12-21

**Summary:**

The common issues lied in:
1. It lacked clear contribution, by combining existing techniques together, and theoretical analysis was missing.
2. Limited generalization across model scales, and the setup and analysis of experiments were questioning.

The authors did not reply to the review comments.

It has to be rejected.

**Reviewer Concerns:**

No rebuttals.

**Reviewer Scores:**

It received 4 reviews with the average score 4.

No discussions for no rebuttals.

---

### Decision · Program_Chairs · 2026-01-26

Reject